



# Predicting abundance and variability of ice nucleating particles in precipitation at the high-altitude observatory Jungfraujoch

Emiliano Stopelli[1], Franz Conen[1], Cindy E. Morris[2], Erik Herrmann[3], Stephan Henne[4], Martin Steinbacher[4], Christine Alewell[1]

[1]: Environmental Geosciences, University of Basel, 4056 Basel, CH
[2]: INRA, UR 0407 Plant Pathology Research Unit, 84143 Montfavet, FR
[3]: PSI, Laboratory of Atmospheric Chemistry, 5232 Villigen, CH
[4]: Empa, Laboratory for Air pollution/Environmental Technology, 8600 Dübendorf, CH

*Correspondence to*: E. Stopelli (emiliano.stopelli@unibas.ch)

**Abstract**

Nucleation of ice affects the properties of clouds and the formation of precipitation. Quantitative data on how ice nucleating particles (INPs) determine the distribution, occurrence and intensity of precipitation are still scarce. INPs active at -8 °C (INPs$_{-8}$) were observed for two years in precipitation samples at the High-Altitude Research Station Jungfraujoch (Switzerland) at 3580 m a.s.l. Several environmental parameters were scanned for their capability to predict the observed abundance and variability of INPs$_{-8}$. Those singularly presenting the best
correlations with observed number of INPs$_{-8}$ (residual fraction of water vapour, wind speed, air temperature, number of particles with diameter larger than 0.5 µm, season and source region of particles) were implemented as potential predictor variables in statistical multiple linear regression models. These models were calibrated with 84 precipitation samples collected during the first year of observations; their predictive power was successively validated on the set of 15 precipitation samples collected during the second year. The model
performing best in calibration and validation explains more than 75% of the whole variability of INPs$_{-8}$ in precipitation and indicates that a high abundance of INPs$_{-8}$ is to be expected whenever high wind speed coincides with air masses having experienced little or no precipitation prior to sampling. Such conditions occur during frontal passages, often accompanied by precipitation. Therefore, the circumstances when INPs$_{-8}$ could be sufficiently abundant to initiate the ice phase in clouds may frequently coincide with meteorological conditions
favourable to the onset of precipitation events.

## 1 Introduction

Ice nucleating particles (INPs) play an essential role in the formation of precipitation on Earth, specifically on
the continents, where most precipitation comes from ice- or mixed-phase clouds (Mülmenstädt et al., 2015). INPs catalyse the first aggregation of water molecules into ice crystals, which progressively grow larger by diffusion of surrounding water vapour and by collision with water droplets and other ice crystals until they reach a sufficient size to precipitate. In the absence of INPs spontaneous freezing would occur only at temperatures below -36 °C (Cantrell and Heymsfield, 2005; Murray et al., 2012).





The scarcity of data about the atmospheric abundance and distribution of INPs prevents a quantitative assessment of their effect on cloud properties, on the development of precipitation and subsequently on climate. Several studies have shown the co-occurrence of INPs from local or faraway sources with precipitation at sites in the Amazon forest (Prenni et al., 2009), in the Sierra Nevada region (Creamean et al., 2013) and at a forested site

in Colorado (Prenni et al., 2013). The prediction of atmospheric concentrations of INPs from more easily accessible parameters would allow a more thorough evaluation of the influence of INPs on clouds and precipitation. This approach merits attention in light of results showing the correlation of specific meteorological and environmental parameters with the abundance of INPs, such as air temperature (Conen et al., 2015), wind speed (Jiang et al., 2014; Jones and Harrison, 2004), relative humidity (Wright et al. 2014), season and

geographical source (Christner et al., 2008), and the abundance of airborne particles of micrometre size (DeMott et al., 2010). In addition, we have recently shown that the abundance of INPs active at moderate supercooling negatively correlates with the amount of water that has been lost from an air mass prior to sampling (Stopelli et al., 2015). All these studies indicate statistical relations between INPs and the mentioned parameters, but each tends to focus predominantly on the role of a single parameter.

Here our objective is to describe and foresee the variations in the concentration of INPs active at -8 °C or warmer (INPs$_{-8}$) in falling precipitation at the high-altitude observatory Jungfraujoch (Swiss Alps, 3580 m a.s.l.) by means of multiple linear regression models. INPs$_{-8}$ are of particular interest since it has been proposed that the ice phase in clouds could be initiated by relatively few INPs$_{-8}$ (10 m$^{-3}$ or less) through the Hallet-Mossop process of riming and ice splintering (Crawford et al., 2012; Mason, 1996). To attain our objective we firstly

identified the strongest predictors for the abundance of INPs$_{-8}$ in precipitation among all the environmental parameters measured at the observatory. Secondly, we implemented these predictors in 3 multiple linear regression models built on the temporal variations in INPs$_{-8}$ occurred during the first year of observations (n = 84). The predictive power of these statistical models was subsequently tested on an independent set of samples from the second year of measurements (n = 15). Prediction of the quantity of INPs$_{-8}$ provides useful means to

guess when and where INPs$_{-8}$ may be sufficiently abundant to impact the formation of the ice phase in clouds and conduce to precipitation.

## 2 Methods

### 2.1 Sample collection and analysis of INPs

Falling snow was collected at the High Altitude Research Station Jungfraujoch in the Swiss Alps (7°59'06'' E, 46°32'51'' N, 3580 m a.s.l.) from December 2012 until October 2014. A total of 106 precipitation samples were collected over these two years, with a median sampling duration of about 2 hours per sample (sampling time between 1.5 and 8 hours), depending on the intensity of the precipitation events. We started sampling campaigns

when the forecasts predicted 2 or more days of precipitation to assure the collection of several samples during each campaign. The station was always inside clouds while sampling, allowing us to collect falling snow as close as possible to where it formed. Samples were collected with a Teflon-coated tin carefully rinsed with ethanol and sterile Milli-Q water to avoid cross-contamination.

Snow samples were analysed for the concentrations of INPs$_{-8}$ directly on site, using the automated drop freeze

apparatus LINDA loaded with 52 tubes containing 100 µl of sample each (Stopelli et al. 2014, Stopelli et al., 2015).





### 2.2 Parameters related to the concentration of INPs

INPs are efficiently removed by precipitating clouds (Stopelli et al., 2015). Therefore, important information on the residual abundance of INPs in rain and snow samples is contained in the value of the residual fraction of

5 water vapour in the sampled air mass $f_V$. Water molecules containing the stable isotope $^{18}O$ have a greater propensity to condense, hence to precipitate, than those containing the more abundant stable isotope $^{16}O$. Consequently, the $^{18}O$:$^{16}O$ ratio (indicated as $\delta$) in an air mass decreases during precipitation. $f_V$ can be easily calculated comparing the isotopic ratio of the initial water vapour content of an air mass at the moment of its formation with the ratio at the moment of sampling according to Rayleigh's fractionation model (IAEA, 2001):

$$\frac{\delta_V}{1000} = \left(1 + \frac{\delta_{V,0}}{1000}\right) \cdot f_V^{\alpha_{L/V}-1} - 1$$

In this study $\delta_V$ is the isotopic ratio of the vapour at Jungfraujoch, calculated from the isotopic ratio of sampled snow, $\delta_{V,0}$ is the modelled isotopic ratio of the vapour in an air mass at the moment of its formation in contact with seawater and $\alpha_{L/V}$ is the fractionation factor from liquid to water along the trajectory of a cloud. Further details on the calculation of these parameters are presented in Stopelli et al., 2015.

Wind speed, air temperature and relative humidity are continuously measured at Jungfraujoch by MeteoSwiss and are stored as 10-minute averages. An optical particle counter (GrimmTM, Dust Monitor 1.108) mounted in series with a heated inlet regularly measures the total number of particles with a dry optical diameter larger than 0.5 μm ($N_{>0.5}$) (Weingartner et al., 1999;WMO/GAW, 2003). To produce robust statistics, it was important to assign a single value of air temperature, wind speed and $N_{>0.5}$ to each snow sample. These parameters had a finer

temporal resolution compared to the measurements of INPs$_{-8}$, therefore they were averaged over the time interval during which each snow sample was collected. To fill gaps due to instrument failures, missing $N_{>0.5}$ values (26 out of 106 samples) were estimated by linear regression from measured $PM_{10}$ concentrations ($R^2$ = 0.40, p < 0.001), which are continuously determined at Jungfraujoch by Empa through beta-attenuation method (Thermo ESM Andersen FH62 IR). Due to the usually low $PM_{10}$ concentrations at Jungfraujoch, data are

aggregated to hourly averages to achieve better signal to noise ratios. In this case, the $PM_{10}$ concentration corresponding to each snow sample was calculated averaging the hourly values including the whole duration of the collection of the sample. Empa also provided hourly values CO and the concentrations of total reactive nitrogen $NO_y$ in the air. The ratio $NO_y \cdot CO^{-1}$ is a common proxy of the age of an air mass, thus it was used as indicator of planetary boundary layer influence and recent land contact of air masses sampled at the observatory

(Paney Deolal et al., 2013; Griffiths et al., 2014). Due to different susceptibility to photochemical transformation in the atmosphere, $NO_y \cdot CO^{-1}$ ratios decrease during transport after being emitted from anthropogenic sources. Therefore, a larger ratio of $NO_y \cdot CO^{-1}$ is associated with a more recent contact of the air masses with land surface. Threshold values in the range 0.002 to 0.008 have been proposed to distinguish between conditions influenced by planetary boundary layer and free tropospheric air masses (Fröhlich et al., 2015; Pandey Deolal et

al., 2013).

Precipitation intensity (mm·h$^{-1}$) was calculated by dividing the water-equivalent volume of precipitation collected in the sampling tin by its horizontal surface and the sampling duration.

Potential regions where air masses could have picked up particles on their way to Jungfraujoch were determined by the analysis of source sensitivities simulated with FLEXPART, a Lagrangian particle dispersion model used

in backward mode (Stohl et al., 2005). FLEXPART was driven with analysed meteorological fields taken from





the ECMWF Integrated Forecasting Systems with a horizontal resolution of 0.2° by 0.2° over the Alpine area and 1° by 1° elsewhere (more details on the specific set-up for JFJ simulations can be found in Brunner et al., 2013). A "source region score" was assigned to each sample, combining information derived from the visual inspection of potential source regions in FLEXPART plots with the prevailing wind direction during sampling.

This categorical parameter was conceived to mirror the potential differences in source quality and source strength of INP populations between northern and southern Europe. Three groups were identified: North, South and mixed/uncertain conditions. *A priori* it was hypothesised that a higher score should be given to samples from Southern Europe, assuming a larger influence of warmer air masses, enriched in larger mineral dust and organic material emissions, also linked to a more prolonged duration of agricultural activities (Kellogg and Griffin, 2006;

Lindemann et al., 1982; Morris et al., 2014). Therefore the larger value should be assigned to events from South, followed by mixed conditions and by events from Northern Europe. The best combination of values was subsequently determined through successive comparisons with the numbers of INPs$_{-8}$ and corresponds to: South = 3; mixed condition = 2, North = 1.

A similar approach was used to insert the "season score" a categorical parameter mirroring the potential effects

of seasonality on the release and abundances of INPs$_{-8}$. *A priori* the highest value was assigned to samples collected in summer, assuming both a larger release of soil and organic material containing INPs, associated with the growth of vegetation, agricultural activity and warmer air masses (Jones and Harrison, 2004; Morris et al., 2014) and a greater chance for INPs to reach the observatory before being removed by precipitation (Conen et al., 2015). In the ranking summer was followed by autumn and spring as intermediate seasons, and finally by

winter. Once this classification was established, the precise values for each class were again determined by comparison with measured INPs$_{-8}$. The best fit with the data was found for the combination: summer = 4; autumn = 3; spring =2; winter =1.

### 2.3 Statistical analyses and modelling

Univariate statistics were carried out with PAST software version 2.17. The R software version 3.0.1 was used to build multiple linear regression models (Hammer et al., 2001; R Team, 2011).

The first step in model building consisted of a preliminary screening of the environmental parameters that had a significant relation with the variability in INPs$_{-8}$. This was done considering both the results of parametric linear regression and Spearman's non-parametric correlation test. Normal distribution of variables is required for

parametric statistics. In particular, the concentrations of INPs$_{-8}$ were approximately log-normally distributed over several orders of magnitude. Therefore, they were $\log_{10}$-transformed to normalise their distribution. This led to the exclusion of 7 of 106 samples with no measurable activity (< 0.21 INPs$_{-8} \cdot$ml$^{-1}$): the arbitrary assignment of small concentrations would have resulted in a large bias when projected on the log scale. Similarly, the number of particles N$_{>0.5}$, precipitation intensity and the ratio NO$_y \cdot$CO$^{-1}$ were $\log_{10}$-transformed to

improve the distribution of their data. Non-parametric correlation was added to draw more robust and stricter conclusions, independent from parameter distributions.

Multiple linear regression models were built on the parameters presenting the best correlations with INPs$_{-8}$. Criteria to build up the models were: (a) to start from the addition of two parameters which we *a priori* suspected could be descriptors of environmental processes impacting INPs$_{-8}$ in different ways, such as proxies for their

production and removal; (b) to add further parameters only if resulting in a significant gain in explained





variability and improved distribution of the residuals; (c) to prefer combinations of parameters weakly correlated among themselves (Table 1), to avoid collinearity.

The normal distribution of independent and dependent variables is not considered necessary for assessing the quality of multiple linear regression models, but it can improve the quality of the assessment. Consequently, the

variables which were log-transformed for univariate statistics were kept transformed also in multiple linear models. The quality of a multiple linear regression model is evaluated by the significance of the whole model as well as of the regression coefficients of each parameter. Particular care was taken in analysing the residuals of the models. All the models presented here fulfilled the conditions of normally distributed residuals, with an average value of zero and no significant trends. Furthermore, we assumed that the parameters could be added in

linear combinations. The correctness of this assumption was verified by the method of partial regression plots of the residuals. Interactions between parameters were tested as well as potential ways for improving the models, but no interaction resulted in significant improvement.

**3 Results and discussion**

**3.1 Model calibration**

The observations used to create the models consist of 84 snow samples with measurable concentrations of $INPs_{-8}$ collected in the Swiss Alps at 3580 m altitude between December 2012 and September 2013. Measured values of $INPs_{-8}$ ranged from the lower limit of detection ($0.21 \cdot ml^{-1}$) to a maximum of $434 \cdot ml^{-1}$. Interestingly, these values

are comparable to, or even greater than those recently found in cloud water samples in Central France at 1465 m altitude (Joly et al., 2014) and well within the range of values and variability observed in precipitation samples collected all around the world (Petters and Wright, 2015).

The best correlations found at Jungfraujoch agree with our current understanding of the factors that can influence the abundance of INPs in the environment (Fig. 1). In particular, the relationships with the remaining fraction of

water vapour $f_V$ and air temperature are coherent with the observation that INPs are rapidly lost by precipitating clouds, hence are more abundant at early stages of precipitation (Stopelli et al., 2015) and that colder air masses tend to be more depleted in $INPs_{-8}$ (Conen et al., 2015). A better fit suggests that $f_V$ is a factor capable of better representing the temporal variability in $INPs_{-8}$ than air temperature. This can be clearly seen for temperatures around 0 °C, at which values of $INPs_{-8}$ cover several orders of magnitude. Whilst $f_V$ is a descriptor of the

cumulative precipitation history of an air mass, air temperature appears more like a local snapshot-value for the activation of $INPs_{-8}$.

Wind speed is a good proxy of the energy and turbulence associated with an air mass, promoting the transport and mixing of airborne particles (Jiang et al., 2014; Jones and Harrison, 2004). This is confirmed by the correlation between wind speed and $\log(N_{>0.5} \cdot m^{-3})$ (Table 1). Wind speed is not correlated to the direction of air

masses, expressed by the source region score, indicating that the local morphology plays a minor role regarding this parameter. Coherently, the correlation between $INPs_{-8}$ and $N_{>0.5}$ suggests that the more particles $N_{>0.5}$ are present in the air, the greater is also the probability of finding a greater abundance of $INPs_{-8}$. This relationship proved significant for INPs active at -15°C or colder (DeMott et al., 2010). Here we show its validity for INPs active at -8 °C measured in precipitation at Jungfraujoch.

$INPs_{-8}$ found in precipitation confirmed the expectations to be more abundant in summer and in air masses coming from Southern Europe. Relative humidity appears as a threshold for the abundance of $INPs_{-8}$ (Wrigth et



al., 2014), with a similar distribution of the data to the one shown by temperature. Therefore, the relationship between the relative humidity and $INPs_{-8}$ may reflect the role of particle processing in the residual abundance of $INPs_{-8}$. This process can be better represented by temperature or $f_V$, thus preference was given to the latter parameters in building multiple linear regression models. $INPs_{-8}$ are not correlated with the intensity of

precipitation, suggesting that different amounts of precipitation can be generated per INP. The ratio $NO_y \cdot CO^{-1}$ presents a relatively low and homogeneous range of values which are related to air masses with slightly recent contact with land surfaces (the most recent threshold value presented in literature for Jungfraujoch is 0.004, -2.4 on log scale, Fröhlich et al., 2015). Nevertheless, the sampling happened always inside precipitating clouds, which suggested the occurrence of the uplift of planetary boundary layer air to the height of the station.

Therefore, it is realistic to speculate that the precipitation collected was generally originating from air masses integrating several source regions and distances before reaching the observatory. Furthermore, the ratio $NO_y \cdot CO^{-1}$ is weakly correlated with $INPs_{-8}$, suggesting that the effect of washout of $INPs_{-8}$ may play a larger role on their abundance than the proximity of the last contact with a land surface.

The parameters presenting the best correlations with $INPs_{-8}$ were successively added into multivariate linear

regression models and the three models predicting the concentrations of $INPs_{-8}$ best are:

$log\ (INPs_{-8} \cdot ml^{-1}) =$

    1)        $2.84 * f_V + 0.02 * wind\ speed\ (km \cdot h^{-1})$ -1.12

    2)        $0.36 * season + 0.02 * wind\ speed\ (km \cdot h^{-1}) + 0.13 * source\ region$-1.39

    3)        $0.02 * wind\ speed\ (km \cdot h^{-1}) + 0.05 * temperature\ (°C) + 0.34 * log(N_{>0.5} \cdot m^{-3}) - 1.54$

They were all capable of describing about 75% of the observed variability for the calibration period (year 1, Table 2) and of reproducing observations equally well, where an apparent seasonal trend with maximum values of $INPs_{-8}$ in summer is recognisable (Fig. 2, upper left panel). Yet, model 1, based on two variables only - $f_V$ and wind speed -, performed slightly better than the other two models, which are based on three parameters. It also provided the smallest maximum absolute error (Table 2). The range of potential concentrations of $INPs_{-8}$ which

can be predicted from model 1 is also the closest to observations. Inserting in the model the smallest, and largest, observed values of $f_V$ and wind speed results in a range of calculated concentrations of 0.11 $INPs_{-8} \cdot ml^{-1}$, and 750 $IN_{-8} \cdot ml^{-1}$. Doing the same with model 2 results in a maximum value of 250 $INPs_{-8} \cdot ml^{-1}$, and with model 3 of 400 $INPs_{-8} \cdot ml^{-1}$, underestimating the range of measured concentrations for at least one event. The observed rapid changes in the abundance of $INPs_{-8}$ may explain the slightly better performance of model 1. Differences in the

concentration of $INPs_{-8}$ of more than 2 orders of magnitude were found not only on a seasonal time scale, but also within the same precipitation event over a couple of hours. The variables "season" and, to a lesser extent, "source region", "temperature" and "$log(N_{>0.5} \cdot m^{-3})$" could not always reproduce such sudden changes, as can be seen from the broader distribution of the dots for these parameters in Fig. 1.

The pattern of residuals over time is almost the same for all three models (Fig. 2, lower left panel). Thus, it is

unlikely to result from random noise and suggests the presence of at least one further driver of the abundance of $INPs_{-8}$ in precipitation. Given the lack of any relationship with precipitation intensity, a likely candidate is the average mass (equivalent liquid volume) of hydrometeors formed by individual INPs. For snow crystals it spans over more than an order of magnitude (Mason, 1957). INPs generating larger hydrometeors, such as those grown through riming, will be diluted in a larger volume of water and result in an overestimation of modelled numbers

of $INPs_{-8}\ ml^{-1}$. Smaller than average ice crystals will do the opposite.



### 3.2 Model validation

We validated the three models with observations from 15 precipitation samples collected between May 2014 and October 2014. The values of the environmental parameters associated with these 15 independent samples (like $f_V$, air temperature and so on) were directly inserted in the equations of the 3 models as derived from the

5 calibration step and used to predict values of $INPs_{-8} \cdot ml^{-1}$.

The observed concentrations of $INPs_{-8}$ during this second period of measurements ranged from $0.21 \cdot ml^{-1}$ to a maximum of $60 \cdot ml^{-1}$. Interestingly, the samples collected in 2014 presented a completely different pattern compared to the previous year of observations (Fig. 2, upper right panel). The lowest concentrations of $INPs_{-8}$ were observed during summer, whilst the highest concentrations occurred in May during a Saharan dust event

and in October when a cold front from Northern Europe reached Jungfraujoch (air temperature = - 16 °C). For the sampling campaigns carried out in June, July and September the local air temperature was relatively warm (between -7 and +3 °C). Still, $f_V$ values were low, between 0.23 and 0.47, suggesting that air masses had already lost substantial parts of their initial water vapour prior to arrival at Jungfraujoch, even if season, source region and local temperature could have been favourable for an abundant residual presence of $INPs_{-8}$.

Thus, models 2 and 3, which are based either on season, source region or air temperature, predicted a smaller variability of INPs than observed and overestimated the low concentrations measured in summer 2014 causing larger residual values (Fig. 2 lower right panel, Table 2). Model 1, based only on two parameters $f_V$ and wind speed, provided the best results in describing the variability of $INPs_{-8}$, also for validation. It produced the lowest absolute errors, less than one log unit (Table 2) and its results more closely matched the observed dynamics and

approached the optimal 1:1 agreement between predicted and measured values over the full range of observed concentrations of $INPs_{-8}$ (Fig. 3).

### 3.3 Fraction of clouds with enough $INPs_{-8}$ to be conducive to the ice phase

A concentration of 10 $INPs_{-8} \cdot m^{-3}$ of air has been observed through in-cloud measurement as a sufficient number

to initiate the ice phase (Crawford et al., 2012; Mason, 1996). This allows calculating a conservative estimate of the fraction of observations where $INPs_{-8}$ could theoretically still have initiated the ice phase in clouds, if it had not already started earlier (note: we sampled already precipitating clouds). Considering the observed median value of 0.25 ml of precipitation from 1 $m^3$ of air (Stopelli et al., 2015), around 40 $INPs_{-8} \cdot ml^{-1}$ should be sufficient to indicate that a precipitation sample originated from a cloud where $INPs_{-8}$ would still have been

numerous enough to induce the ice phase. Referring to the full set of samples used for calibration and validation, such a concentration was found or exceeded in 12 (10 %) of 106 samples.

### 3.4 Open question: source and sink effects

It remains an open question whether the seasonal change in the source strength of $INPs_{-8}$ or whether their

selective removal is more relevant to their final concentration in precipitation. Still, we know that several surfaces on Earth host organisms with ice nucleating activity or producing ice nucleating molecules, which may equally contribute to the potential release of airborne $INPs_{-8}$: oceans (De Mott et al., 2015; Schnell and Vali, 1975; Wilson et al., 2015), soils (Conen et al., 2011; Fröhlich-Nowoisky et al., 2015; O'Sullivan et al., 2015; Pummer et al., 2015), crops and forests (Huffman et al., 2013; Lindemann et al., 1982; Lindow et al., 1978;

Morris et al., 2014; Pöschl et al., 2010; Prenni et al., 2013), leaf litter (Monteil et al., 2012; Schnell and Vali, 1972; Schnell and Vali, 1973), lichens (Moffett et al., 2015) and freshwater (Morris et al., 2010). In addition to





that, the good correlations between INPs$_{-8}$ and $f_V$ and between INPs$_{-8}$ and wind speed presented in this study hold true not only over a year of observations, but also within several multi-day sampling campaigns. Therefore it is possible to imagine that, independent from a more or less constant and widespread reservoir of INPs$_{-8}$, it is the combination of the energy of an air mass with the amount of precipitation generated by this air mass that

determine together the residual abundance of INPs$_{-8}$ in precipitation samples. In this perspective, the roles of source region and seasonality may reflect the potential for an air mass to reach Jungfraujoch with a lot of particles and little prior precipitation rather than a different background number of airborne INPs$_{-8}$.

**4 Conclusions**

Our investigation indicates that a large abundance of INPs$_{-8}$ in precipitation at Jungfraujoch is present whenever there is a coincidence of high wind speed and moist air mass with little or no prior precipitation. Furthermore, we have estimated that for 10 % of our precipitation samples there would have been enough INPs$_{-8}$ to initiate the ice phase in clouds. Based on the results of the present study, INPs active at moderate supercooling are expected to

15 play a role whenever conditions of high wind speed and first precipitation from air masses prevail. These conditions can be met when an air mass is suddenly forced to rise, for instance at the boundary of a front or due to thermal updrafts or when crossing a mountain ridge. Specifically during the passage of a cold front, gusty winds promote the uptake of particles and the first clouds that form will still retain a large fraction of the initial water vapour of the warm air mass (Gayet et al., 2009; Wright et al., 2014). Simultaneously, physical conditions

along a cold front are favourable for cloud formation. Therefore, frequent systematic coincidences of high numbers of INPs with meteorological conditions conducive to precipitation may be expected. Due to this co-occurrence, the potential impact on precipitation by INPs active at slight supercooling - such as INPs of biological origin - may be larger than previously estimated. Their role in the water cycle might therefore best be studied under such conditions.





**Author contributions**

E.S. and F.C. did the field measurements at Jungfraujoch on the concentrations of INPs, analysed data, did statistical modelling and wrote the manuscript. E.H., S.H. and M.S. respectively provided data and support for the interpretation of $N_{>0.5}$, FLEXPART modelling, $PM_{10}$ and trace gases and contributed to writing the paper.

C.M. and C.A. provided strong conceptual frameworks and contributed to writing the paper.

**Acknowledgements**

We thank the International Foundation for High Alpine Research Station Jungfraujoch and Gornergrat (HFSJG) for making it possible for us to conduct our measurements at Jungfraujoch. Urs and Maria Otz, Martin and Joan

Fischer provided helpful support during field activity. Corinne Baudinot measured the abundance of INPs in snow samples during the second year of observations. Dr Thomas Kuhn and Mark Rollog analysed the stable isotope ratio in our snow water samples. We thank MeteoSwiss for providing data on meteorology at Jungfraujoch. The work described here was supported by the Swiss National Science Foundation (SNF) through grant no 200021_140228 and 200020_159194. Measurements of total solid particles were performed by Paul

Scherrer Institute in the framework of the Global Atmospheric Watch (GAW) programme funded by MeteoSwiss with further support provided by the European FP7 project BACCHUS (grant agreement no. 49603445).

The authors declare no competing financial interests.



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



**Table 1:** Pairwise correlations among predictors for the concentrations of INPs$_{-8}$. The upper panel indicates the pairwise Spearman's $r_s$ correlation coefficient, the lower panel its statistical significance (*,**,*** stand for a probability smaller than 0.05, 0.01 and 0.001 respectively).

| | $f_V$ | wind speed | temperature | $\log(N_{>0.5} \cdot m^{-3})$ | season | source region | relative humidity | $\log(mm \cdot h^{-1})$ | $\log(NO_y \cdot CO^{-1})$ |
|---|---|---|---|---|---|---|---|---|---|
| $f_V$ | | 0.10 | 0.88 | 0.33 | 0.89 | 0.42 | 0.72 | 0.07 | 0.13 |
| wind speed | | | 0.02 | 0.21 | -0.04 | 0.17 | 0.04 | 0.08 | 0.31 |
| temperature | *** | | | 0.18 | 0.85 | 0.58 | 0.70 | -0.03 | -0.28 |
| $\log(N_{>0.5} \cdot m^{-3})$ | ** | * | * | | 0.35 | 0.32 | 0.28 | -0.11 | 0.37 |
| season | *** | | *** | *** | | 0.38 | 0.77 | -0.04 | 0.15 |
| source region | *** | | *** | ** | *** | | 0.42 | -0.12 | -0.23 |
| relative humidity | *** | | *** | ** | *** | *** | | -0.12 | 0.23 |
| $\log(mm \cdot h^{-1})$ | | | | | | | | | 0.24 |
| $\log(NO_y \cdot CO^{-1})$ | | * | * | ** | | * | * | * | |





**Table 2:** Summary of the main statistics employed to evaluate the quality of the models. "$R^2_{adj}$" is the fraction of the observed variance reproduced by a model. It is adjusted to account for the number of variables and samples considered. All models, parameters, and constants are highly significant ($p < 0.001$), except the parameter "source region" ($p = 0.06$). "$\beta^*$" is the value of the standardised regression coefficients, expressing the relative importance of each parameter in a model. The "residuals" column shows the median residual value "med" and the maximum absolute residual as maximum estimation error "ABS" (the corresponding values on linear scale are shown in brackets). "MSE" is the value of the mean squared error.

| | Calibration | | | Validation |
|---|---|---|---|---|
| | $R^2_{adj}$ | $\beta^*$ | residuals | residuals |
| 1 | **0.76** | $f_V$: **0.62** <br> wind speed: **0.47** | Med: -0.04 (0.9) <br> ABS: 1.02 (10.4) <br> MSE: 0.16 | Med: -0.08 (0.8) <br> ABS: 0.87 (7.4) <br> MSE: 0.25 |
| 2 | **0.73** | season: **0.52** <br> wind speed: **0.52** <br> source region: **0.12** | Med: 0.00 (1.0) <br> ABS: 1.45 (28.4) <br> MSE: 0.17 | Med: -0.25 (0.6) <br> ABS: 1.42 (26.1) <br> MSE: 0.56 |
| 3 | **0.74** | wind speed: **0.49** <br> temperature: **0.48** <br> $\log (N_{>0.5} \cdot m^{-3})$: **0.26** | Med: -0.02 (0.9) <br> ABS: 1.22 (16.5) <br> MSE: 0.17 | Med: -0.33 (0.5) <br> ABS: 1.13 (13.5) <br> MSE: 0.43 |



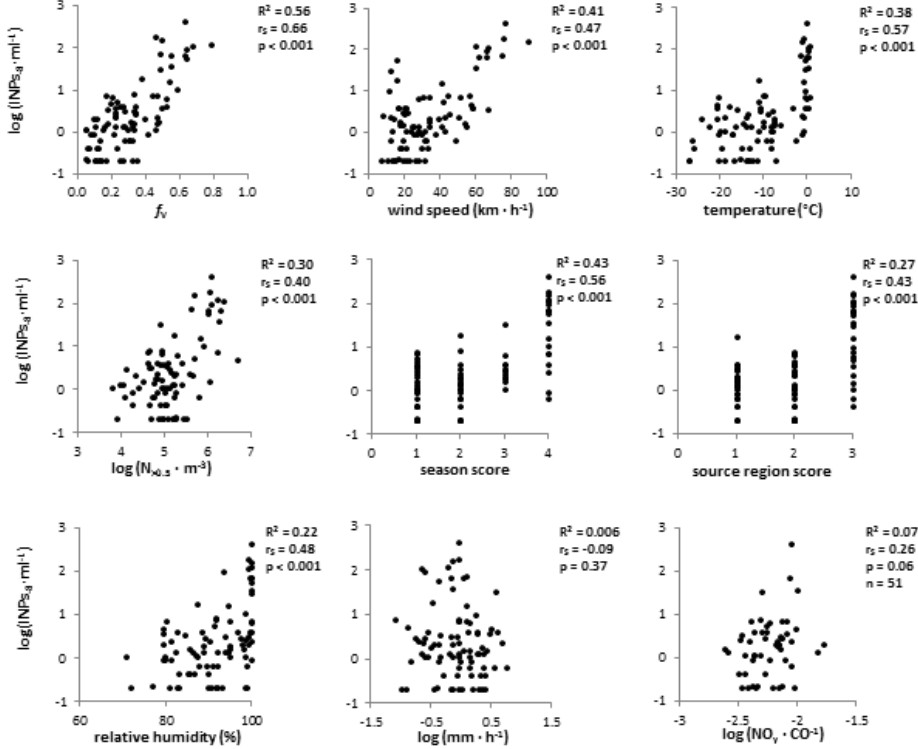

**Fig. 1:** Relationships between INPs$_{-8}$·ml$^{-1}$ and singular environmental parameters. Shown are R$^2$ values, r$_s$, the coefficient of Spearman's correlation, and its probability p. The number of samples (n) was always 84, exception for the ratio NO$_y$·CO$^{-1}$ (n = 51) due to missing trace gases. $f_V$ indicates the remaining fraction of water vapour in a precipitating air mass. In the panel "season score" 1 is for winter, 2 for spring, 3 for autumn and 4 for summer. In the panel "source region score" 1 is for Northern Europe, 2 for mixed conditions and 3 for Southern Europe.





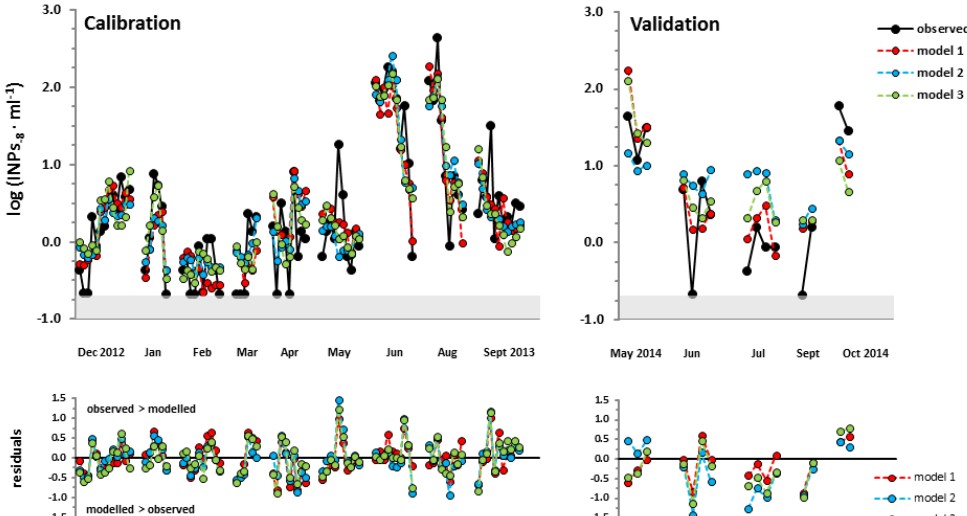

**Fig. 2:** Comparison of observed concentrations of INPs$_{-8}$ with the values predicted by models (absolute values in upper panels, residuals in lower panels). On the left side the results of model calibration are presented, on the right side those for the validation of the models. The grey area in the upper panels indicates the range below the detection limit of our observation method. Time proceeds from left to right, intervals are not to scale, dots belonging to the same sampling campaign are connected by dashed lines.





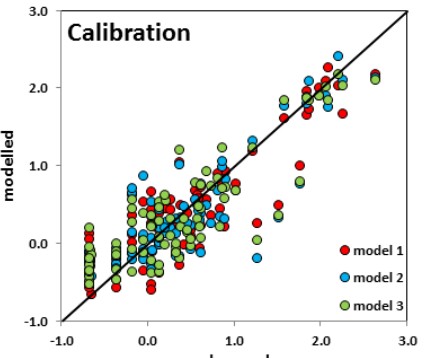
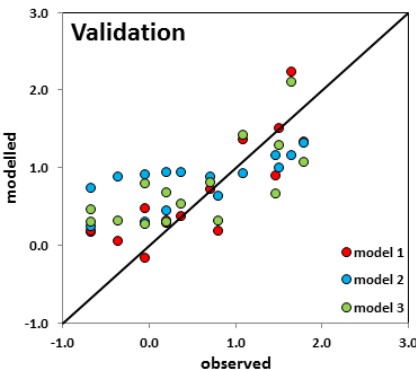

**Fig. 3:** Fitting of modelled concentrations of INPs$_{-8}$ to the observed values, expressed on the log scale. The black line represents the perfect 1:1 correspondence. Specifically for the validation, it is possible to observe how model 2 and 3 tend to overestimate the lowest concentrations of INPs$_{-8}$ measured during summer 2014.