# Peer review of "Predicting abundance and variability of ice nucleating particles in precipitation at the high-altitude observatory Jungfraujoch"

_Atmospheric Chemistry and Physics, 2016_

## Referee Comment (RC1) · Anonymous Referee #1 · 14 Mar 2016

Stopelli and co-authors present a series of ice nucleating particle concentrations at -8 $^{\circ}$C in precipitation (INPs-8) from over one hundred samples taken at the Jungfraujoch research station in Switzerland. The results, which show concentrations ranging from as high as 434 INPs-8 ml-1 to below the limit of detection, are then compared with the range of 'standard' meteorological, aerosol and chemical observations at this site. The results from the first $\sim$80 % of the observations are then used to create models to predict the INP concentration, with the last 20 % of observations used as a model validation. The models provided suggest that the INP concentration in precipitation is potentially driven by wind speed, temperature, aerosol concentration, the season and source region, or the precipitation history of the air mass.

The manuscript is in my opinion ideal for publication in Atmospheric Chemistry and Physics, and is also in a good condition. I have a number of minor comments and some technical corrections (listed below), however it must be noted that based on my personal background an in-depth critique of the statistical model part of the paper cannot be given.

Minor Comments:

Introduction: The introduction explains the importance of INP's for cloud processes and precipitation formation and then talks about measurements of INP's and why they are relevant at high temperatures. However, I missed any discussion of how the observations made, specifically of INP's in collected precipitation water, relates to the cloud processes themselves. There are various processes that would increase or reduce the concentration of INP's in collected water; these and their possible effects on the measurements should be discussed.

Section 2.1.: How were the settling of particles and losses/experimental artefacts prevented/accounted for? What is the instrumental background?

Section 2.1.: Is there any other details of the precipitation water? For example, particle concentration and size distributions could some indication (rather approximate) of the number of primary ice particles per ml of water. Some correlation with the concentration of particles could be expected.

Section 2.2.: This section launches straight in to a listing of the considered parameters. Some short introduction might be helpful to give some context.

Section 2.2.: For the "source region score" and "season score", the text (specifically page 4 line 20) implies that numbers other than 1, 2, 3 and 4 were also considered. If so, it might be worth explicitly stating so and also stating the range of numbers considered.

Section 2.3.: Does the exclusion of 'zero' samples not cause a bias towards higher

values? Did you consider trying the same fitting process with, for example, random concentrations 0 <> 0.21 INPs-8 ml-1?

Section 2.3.: For the multiple linear regressions, were many other models considered? What about other possibly intuitive combinations? Maybe some of the less favourable models could be included in Table 2.

Section 2.3., page 5, lines 10-12: You should consider providing more detail for the concepts in both sentences for the less statistically-minded reader.

Section 3.1., page 6, line 12: it would make it clearer for the reader if you explicitly state why the weak correlation suggests the washout of INP is more important.

Section 3.2.: Could the absence of one of the seasons in the validation data have any effect on the performance of second model?

Section 3.3.: I felt this section ended a little prematurely. Maybe the implications of this could be discussed. Would this imply that the 10 L-1 concentration requirement is too high, or is there some other factor that means the observations are below this level?

Section 3.4.: I am not sure what the extensive listing of various INP sources and literature adds to the argument. Consider removal or replacement with an appropriate review paper.

Section 3.4., page 8, line 2: Please explain what the several multi-day sampling campaigns were. Is this the validation data?

Section 4.: The conclusions is, I feel, too brief. I came out of this feeling that I know why you did the work, but not why I should care about the result. Please add a summary at the beginning, and provide some conclusions with regards to the models. The second half of this section (lines 15-24) reads much more like it should be at the end of the introduction – either move it there or make it more relevant to the work and results presented. The only real conclusion presented is not very impressive (line 14) – the main outcomes of the work need to be better stated and some implications.

Table 2: Consider adding the actual fit parameters to the table. I'm not sure that you should include the linear-scale values, as the proportionality of these depends on where you are in the log scale - taking the log-residual of something at $10^3$ and then re-linearising relative to $10^0$ it does not give the actual linear value of the residual.

All figures: please add error bars.

Figure 1: For the classified data it might be helpful to add the number of data points per class somewhere. Could the number of datapoints in any class be high enough to cause any potential bias to the fits? Is the data from year 2 included? If so show it with a different symbol (add it if not shown). This would help with the discussion in section 3.2.

Figure 2/3: suggest adjusting the symbols such that the paper can be printed and understood in black and white.

Figure 3: This figure isn't really discussed in the text. I would consider either expanding its presence in the text or removing it.

Technical Comments:

Page 3, line27. Suggest insert "of" before CO, and changing "values" to concentrations, and deleting "the concentrations of" from the middle of the sentence: "...hourly concentrations of CO and total reactive...".

Page 4, line 14: insert a comma after "season score".

Page 5, line 41: Citation for Wright et al is misspelt.

Page 6, line 33: Delete "the dots for".

Page 7, line 15: Please re-write the sentence so that the paragraph does not start with the word "Thus".

Page 7, line 18: Delete ", also for validation".

Page 7, line 37: "DeMott", not "De Mott" (please check elsewhere as well).

---

## Referee Comment (RC2) · Anonymous Referee #2 · 9 Apr 2016

Stopelli et al. analyze a two year time series of ice nucleating particles collected from falling precipitation at the Jungfraujoch high altitude observatory. The data are analyzed using simple correlation and multi-linear regression analysis.

I read the paper with great interest. Understanding atmospheric variability of 'warm' ice nucleating particles is important and the factors that control their abundance are poorly understood. This paper presents some new insights on the subject. However, the multi-linear regression models do not help the community to glean widely generalizable conclusions. They are location specific. Their ability to explain the temporal variation in the data does not rise beyond an order of magnitude estimate.

The conclusions are overstated. This work shows that there is some statistical cor-
rleation between INP @-8C in precipitation and temperature, fv, wind speed, source region, N500, and source region. Understanding the uncertainties in the data, the physical mechanisms driving these correlations, and the covariance matrix between the identified parameters is needed. Any claim about the role of INP in precipitation and impact on the water cycle is not supported by this work needs to be removed from the paper. The authors need to more clearly articulate the utility and limitations of the proposed statistical models.

Notwithstanding a report on the inferred correlations merits publication to guide the experimental design of future studies.

Comments: The dynamic range of the measured signal log(INP @ -8C) is ∼2.5 orders of magnitude, ranging from ∼-0.5 to ∼2 in Fig. 2. The majority of data fluctuate between -0.5 and 0.5, and so do the residuals of the model. The statistical model is successful in identifying the high INP episodes Jun/Aug 2013 and May 2014. Other fluctuations seem to be mostly noise in either the data, the model, or both. I have several question regarding Fig. 2 and the model. (1) What is the uncertainty in the concentration (temperature) data? Were frequent pure water tests run to determine the measurement noise? (2) INP spectra can be steep functions of temperature. How do small errors in temperature propagate into INP @-8C? (3) It seems that the calibration time-series for the model is at higher resolution than the data? (Trying to count green and black dots and get different numbers). (4) How does measurement noise, in principle, affect the multi linear regression analysis? Residuals can be interpreted as either measurement error or insufficient number or assumed physical relationship of the proposed explanatory variables. How can these be separated?

The highest INP were observed for very high wind speeds (60-80 km h-1). Were precautions taken that resuspended snow and wind-blown aerosol did not contaminate the measurements?

Even if the source is local, how could local vs. long-range sources be distinguished?

It is difficult to foresee all the possible cross-correlations. One of the strongest predictors for high INP seems to be temperature (even though the $R^2$ is low). Values of log(INP) > 1 only under the condition that temperature is at 0C. Thus the ability of the model to guess the high INP events is to guess when it is warm, which correlates with season, source region, RH, N>500. Assigning explanatory power to any of the variables in terms of a physical mechanism is likely erroneous.

It is still unclear to this referee what the potential application of the statistical would be. A parameterization for a model? Improved process level understanding? Neither approach would seem promising to me without much more analysis. Perhaps the authors have another use in mind?

Relating INP in precipitation to INP in air, and INP in air feeding the cloud, is problematic due to a wide range of known artifacts. These include capture of INP by falling rain snow and chemical or physical modification of INP by the precipitation. None of these issues are acknowledged in the manuscript. Even if the explanatory power of the model were perfect and the intrinsic measurement error of LINDA was zero, what relevance do INP @-8C measured in precipitation have to clouds over Jungfraujoch?

The discussion in Section 3.3 should be removed since they discuss very specific cloud types, updrafts and meteorological conditions. It is entirely unclear to what extent the INP @-8C at Jungfraujoch have any role in precipitation without considering the larger scale meteorological forcing, including liquid water content, cloud top height, updraft velocity, the INP spectrum at warmer and colder temperatures, the efficiency of the competing warm rain process, and the cumulative ice phase history, including possible seeding by wind-blown ice crystals along the storm trajectory, including all mountains upwind. No information can be gleaned from this work about any of these processes.

---

## Author Response (AR1)

We thank the Referees for their constructive comments on our paper and for finding our manuscript interesting for publication. Their suggestions helped us to elaborate more clearly the issues presented in the manuscript and to improve the quality of the work. Here is a point-by-point reply to their proposed corrections. We merged the reply to the Referees into a unique file to improve its clarity and the level of detail.

*Referees' comments are in italic font,* authors' response in blue normal font, changes in the manuscript are in blue underlined and refer to the document published on ACPD. To facilitate the discussion, we have numbered the comments.
* * *
**Referee #1:**

*Stopelli and co-authors present a series of ice nucleating particle concentrations at -8 °C in precipitation (INPs$_{-8}$) from over one hundred samples taken at the Jungfraujoch Research station in Switzerland. The results, which show concentrations ranging from as high as 434 INPs$_{-8}$ ml$^{-1}$ to below the limit of detection, are then compared with the range of 'standard' meteorological, aerosol and chemical observations at this site. The results from the first 80 % of the observations are then used to create models to predict the INP concentration, with the last 20 % of observations used as a model validation. The models provided suggest that the INP concentration in precipitation is potentially driven by wind speed, temperature, aerosol concentration, the season and source region, or the precipitation history of the air mass.*

*The manuscript is in my opinion ideal for publication in Atmospheric Chemistry and Physics, and is also in a good condition. I have a number of minor comments and some technical corrections (listed below), however it must be noted that based on my personal background an in-depth critique of the statistical model part of the paper cannot be given.*

*Minor Comments:*

**(1)** *Introduction: The introduction explains the importance of INPs for cloud processes and precipitation formation and then talks about measurements of INPs and why they are relevant at high temperatures. However, I missed any discussion of how the observations made, specifically of INPs in collected precipitation water, relates to the cloud processes themselves. There are various processes that would increase or reduce the concentration of INP's in collected water; these and their possible effects on the measurements should be discussed.*

The connection between precipitation and cloud processes is particularly important and merits to be discussed in the manuscript. Impaction scavenging of aerosols can contribute to the removal of airborne INPs, increasing their abundance in precipitation. Crystal growth by vapour deposition and riming (if rime does not contain INPs) can on the contrary dilute INPs in precipitation. A possible improvement for similar top-down studies would be to couple measurements of airborne INPs$_{-8}$ with their abundance in precipitation and to find proxies for assessing the scavenging and riming efficiency. Concerning this manuscript, it has to be kept in mind that we were inside precipitating clouds, therefore as close as possible to the formation of precipitation. Furthermore, we generally mention INPs$_{-8}$ as "removed by precipitation" (coherently with Stopelli et al., 2015), which includes both mechanisms of active (nucleation) and passive (scavenging) removal. On page 6, line 37 we also indicate that important missing information is constituted by the "average mass (equivalent liquid volume) of hydrometeors formed by individual INPs". We dedicated a new paragraph to better clarify this issue at the end of section 2.1: "The collection of precipitation allows the sampling of INPs which either formed precipitating ice particles or were scavenged by precipitation. It is

difficult to distinguish between these contributions in field studies, where scavenging, riming and crystal growth by vapour deposition can alter the abundance of INPs in precipitation. Nevertheless, the station was always inside clouds while sampling, allowing us to collect falling snow as close as possible to where it formed. Furthermore, precipitation was immediately analysed, in order to minimise the chance for biases due to artefacts like the production (f.i. the release in solution of INPs or cellular multiplication) and the loss (f.i. settling or increased molecular weakness of biological INPs detached from mineral and soil dust) of $INPs_{-8}$".

**(2)** *Section 2.1.: How were the settling of particles and losses/experimental artefacts prevented/accounted for? What is the instrumental background?*

A detailed description of the blank preparation is provided in comment 1 Referee #2 and has been inserted in the manuscript at the end of section 2.1. Briefly, out of 29 blanks analysed only 2 had measurable $INPs_{-8}$ in concentrations of $0.11 \cdot ml^{-1}$ (the description of the instrument is fully reported by Stopelli et al., 2014). Settling of particles was prevented by gently shaking the sample prior to dividing it rapidly into aliquots. INPs active at moderate supercooling can increase or decrease, specifically when in solution, either due to cellular multiplication, release in solution of molecules, settling or even increased molecular weakness when biological residues detach from soil. We think that such behavior cannot be easily predicted and is sample-specific. The important aspect of this study is that we analysed the sample as soon as possible, trying to reduce as much as possible the impact any of the above mentioned artefacts. This is addressed at the end of section 2.1 in the revised version of the manuscript when discussing about sampling precipitation according to comment 1 Referee #1.

**(3)** *Section 2.1.: Is there any other details of the precipitation water? For example, particle concentration and size distributions could some indication (rather approximate) of the number of primary ice particles per ml of water. Some correlation with the concentration of particles could be expected.*

We do not have such information. Particle numbers in precipitation could be determined by flow cytometry. In practice, counting would probably be limited to larger particles. It could be approximate because INPs active at moderate supercooling can be constituted of cells, but also of cellular fragments and molecules, either freely floating or attached to soil and mineral dust particles of several dimensions. Furthermore, the number of particles in precipitation may be biased by the same scavenging/riming processes as it can happen for INPs, as discussed in comment 1 Referee #1.

**(4)** *Section 2.2.: This section launches straight in to a listing of the considered parameters. Some short introduction might be helpful to give some context.*

At the beginning of section 2.2 this sentence has been added to provide some context: "To analyse and understand more on the factors responsible for the variability of $INPs_{-8}$ in precipitation, several environmental parameters were considered in relation with the number of $INPs_{-8}$".

**(5)** *Section 2.2.: For the "source region score" and "season score", the text (specifically page 4 line 20) implies that numbers other than 1, 2, 3 and 4 were also considered. If so, it might be worth explicitly stating so and also stating the range of numbers considered.*

Yes, based on the *a priori* ranking of seasons and source regions several combinations of numbers were considered. To state it better, line 10 page 4 was changed into: "Therefore *a priori* the larger value should be assigned to events from South, followed by mixed conditions and by events from Northern Europe. Several combinations of values ranging from 1 to 3 (f.i. 2-1.5-1; 3-2-1,…) were tested and the best combination of values was determined through comparisons with the numbers of $\log(INPs_{-8} \cdot ml^{-1})$. It corresponds to: South = 3; mixed condition = 2, North = 1". Similarly, line 20 page 4 was changed into: "Once this *a priori* classification was established, the precise values for each class were again determined by comparing different possible combinations of numbers from 1 to 9 (f.i. 3-2-2-1; 9-6-3-1; 4-3-2-1,…) with measured values of $\log(INPs_{-8} \cdot ml^{-1})$. The best fit with the data was found for the combination: summer = 4; autumn = 3; spring = 2; winter = 1."

**(6)** *Section 2.3.: Does the exclusion of 'zero' samples not cause a bias towards higher values? Did you consider trying the same fitting process with, for example, random concentrations 0 <> 0.21 INPs$_{-8}$ ml$^{-1}$?*

Deciding to exclude the "zero" samples from the fitting process results in a poor constraint for the model at low concentrations, where error associated to counting frozen tubes is larger, but not necessarily in a bias towards larger values. The difference it introduces is that values above the detection limit have a weight in the fitting process and in evaluating the performance of a model. On the contrary, giving "zero" samples a value by randomly assigning a small concentration would have required a subjective judgement. Specifically, we would have had to subjectively decide on the shape of the random distribution >0 <0.21, which on a log scale can be relevant. To avoid any doubts about the objectivity in the model building process, we decided to accept the poor constraint at the very low end of INP concentrations.

**(7)** *Section 2.3.: For the multiple linear regressions, were many other models considered? What about other possibly intuitive combinations? Maybe some of the less favourable models could be included in Table 2.*

Yes, many other models have been considered following the methodological approach described in section 2.3. Still, for the sake of keeping information focused, we would rather exclude them from this paper.

**(8)** *Section 2.3., page 5, lines 10-12: You should consider providing more detail for the concepts in both sentences for the less statistically-minded reader.*

At the end of section 2.3 more details were added to clarify the statistical approach. The end of this paragraph now reads as: "Given an ideal model $y \sim x_1 + x_2$ ($x_1$ and $x_2$ used to derive the dependent variable y) it is possible to test whether f.i. $x_2$ is linearly linked to y. To do that,

the residuals of the regression of y with $x_2$ are plotted against the residuals of the regression of $x_1$ with $x_2$. A linear distribution of the residuals confirms the correctness of the linear relationship between $x_2$ and y. On the contrary, the presence of a different trend implies a different relationship between $x_2$ and y, like, for example, a quadratic one. Interactions between independent variables were tested as well as potential ways to improve the models. This means that the additional factor $x_1 \cdot x_2$ was inserted in a model to test whether the effect of the independent variable $x_1$ (or $x_2$) on the dependent variable y changes according to different levels of the other independent variable $x_2$ (or $x_1$). No interaction we tested resulted in a significant improvement of the models."

**(9)** *Section 3.1., page 6, line 12: it would make it clearer for the reader if you explicitly state why the weak correlation suggests the washout of INP is more important.*

We changed this sentence on page 6 line 12, which now reads as: "Furthermore, the ratio $NO_y \cdot CO^{-1}$ is positively correlated with wind speed and $N_{>0.5}$ (Table 1), suggesting that at high wind speed clouds at Jungfraujoch may be charged with particles taken up during recent contact of the air mass with a land surface. This suggestion could explain larger numbers of $INPs_{-8}$ in precipitation at high $NO_y \cdot CO^{-1}$ ratios (Fig. 1)".

Likely "source" and "sink" processes are further discussed in the revised version of section 3.3 (the former section 3.4).

**(10)** *Section 3.2.: Could the absence of one of the seasons in the validation data have any effect on the performance of second model?*

All models performed similarly in the validation step. We cannot exclude that the "season" parameter in model-2 would have given it an advantage over other models in a validation spanning all four seasons of the second year of observations. Still, it appears as unlikely since we expect that in winter the seasonal score is 1, the temperature is generally low and $f_V$ is also low, therefore all models should perform pretty much the same. Unique in the year 2014 was the sampling in summer of precipitation from relatively warm air masses that, according to the isotopic values, had already lost considerable amount of water prior to arrival at Jungfraujoch.

**(11)** *Section 3.3.: I felt this section ended a little prematurely. Maybe the implications of this could be discussed. Would this imply that the 10 $L^{-1}$ concentration requirement is too high, or is there some other factor that means the observations are below this level?*

The former section 3.3 has been removed from the manuscript according to comment 10 by Referee #2.

**(12)** *Section 3.4.: I am not sure what the extensive listing of various INP sources and literature adds to the argument. Consider removal or replacement with an appropriate review paper.*

We removed the literature and substituted it with this sentence in the revised version of section 3.3, including the review paper of Després et al., 2012 (added in the reference list as well): "To different degrees, surfaces on Earth, like oceans, forests, crops, soils, freshwaters, snow packs, host organisms with ice nucleating activity (Després et al., 2012) and may contribute to the airborne population of INPs$_{-8}$.".

**(13)** *Section 3.4., page 8, line 2: Please explain what the several multi-day sampling campaigns were. Is this the validation data?*

Every sampling campaign lasted several days and we observed that the relationship between INPs$_{-8}$ and $f_V$ or wind speed holds not only for the whole set of data but also in several cases within the data belonging to the same sampling campaign. Section 3.4 has been changed into section 3.3 and the sentence eliminated in the revised version of the manuscript, as reported in comment 8 Referee #2.

**(14)** *Section 4.: The conclusions is, I feel, too brief. I came out of this feeling that I know why you did the work, but not why I should care about the result. Please add a summary at the beginning, and provide some conclusions with regards to the models. The second half of this section (lines 15-24) reads much more like it should be at the end of the introduction – either move it there or make it more relevant to the work and results presented. The only real conclusion presented is not very impressive (line 14) – the main outcomes of the work need to be better stated and some implications.*

We added a summary of the work (page 8 line 11): "Large variability in the abundance of INPs$_{-8}$ has been found in over a hundred precipitation samples collected at Jungfraujoch (3580 m a.s.l.), with values ranging from 0.21 INPs$_{-8}$·ml$^{-1}$ to 434 INPs$_{-8}$·ml$^{-1}$. Strikingly, with simple multiple linear regression models based on some easily measurable environmental parameters it was possible to describe and predict up to 75 % of the observed variability in INPs$_{-8}$. All these models indicate that the variability of INPs$_{-8}$ is determined by the interaction of source and sink processes, such as the potential for air masses to pick up and transport INPs$_{-8}$ and for INPs$_{-8}$ to be removed by precipitation".

The sentence starting at page 8 line 14 now reads: "Based on the results of the present study, INPs active at moderate supercooling are expected to be abundant whenever high wind speed coincides with first (initial) precipitation from an air mass.". Page 8 line 21 now reads as: "Due to this frequent co-occurrence,…"

The former section 3.4 (3.3 in the revised version) has been changed according to comment 8 Referee #2 and should provide a better bridge with the conclusions. As a generalizable conclusion, we expect large abundance of INPs$_{-8}$ at the passage of a front, which also represents the best context to concentrate sampling effort to study the impact of INPs$_{-8}$ on precipitation.

**(15)** *Table 2: Consider adding the actual fit parameters to the table. I'm not sure that you should include the linear-scale values, as the proportionality of these depends on where you are in the log scale – taking the log-residual of something at 10^3 and then re-linearising relative to 10^0 it does not give the actual linear value of the residual.*

All statistics for the analysis of the residuals have been done on the log-transformed values. In brackets we wanted to report the corresponding factor of error estimate on linear scale, not the median INPs$_{-8}$·ml$^{-1}$ residual value. That is to say, if a median residual is 1 on the log$_{10}$ scale, we can expect up to 1 order of magnitude error on a linear scale. To make the concept clearer, we modified the caption of table 2 as follows: "…The "residuals" column shows the median residual value "med" and the maximum absolute residual as maximum estimation error "ABS" (the corresponding factors of error estimate on linear scale are shown in brackets)."

**(16)** *All figures: please add error bars.*

This is an important suggestion, also because calculations of errors are rarely reported. In all figures we added errors bars associated to the concentrations of INPs$_{-8}$, coherently also with comment 1 and 2 Referee #2. Error bars have been cautiously calculated propagating the error associated with the method of droplet freezing (based on the error in counting frozen and unfrozen tubes, assuming a Poisson distribution) into the error associated with the temperature measurement.

To maintain the readability of figures, errors bars are shown close to the graphs for concentrations of 1, 10, 100, 1000 INPs$_{-8}$·ml$^{-1}$ and this has been indicated in the figure captions.

A description of the calculation of errors has been introduced in section 2.1: "Error bars for values of 1, 10, 100, 1000 INPs$_{-8}$·ml$^{-1}$ are shown in the figures of the article. Confidence intervals were calculated as errors in counting frozen tubes, following Poisson's distribution (depending on the number of frozen tubes, they account for 30 % to 50 % increase or decrease of the calculated concentrations of INPs$_{-8}$). These intervals were propagated into the uncertainty associated with the maximum error in the determination of the freezing temperature of the tubes of ± 0.2 °C (assuming a doubling of INPs per °C of decrease in the freezing temperature, an error of 0.2 °C accounts for a change in 14 % of the measured concentrations) to provide more cautious confidence intervals".

**(17)** *Figure 1: For the classified data it might be helpful to add the number of data points per class somewhere. Could the number of data points in any class be high enough to cause any potential bias to the fits? Is the data from year 2 included? If so show it with a different symbol (add it if not shown). This would help with the discussion in section 3.2.*

Data from year 2 were formerly not included since the correlations presented are based only on data from year 1. Green squares have been added to represent data from year 2 to improve the discussion in section 3.2, but were not used for the correlations since they are exclusively employed to validate the data. Reference to figure 1 has been introduced in the section 3.2, while the caption of figure 1 has been adapted to the modifications and now reads: "Relationships between INPs$_{-8}$·ml$^{-1}$ and singular environmental parameters for the first year of observations (black dots, n = 84, exception for the ratio NO$_y$·CO$^{-1}$ with n = 51 due to missing data of trace gases). Shown are R$^2$ values, the coefficient of Spearman's correlation r$_s$ and its probability p$_s$ calculated for the data belonging to the first year of observations. $f_V$ indicates the remaining fraction of water vapour in a precipitating air mass. In the panel "season score" 1 is for winter, 2 for spring, 3 for autumn and 4 for summer. In the panel "source region score" 1 is for Northern Europe, 2 for mixed conditions and 3 for Southern Europe. For both "season score" and "source region", values of Kruskall-Wallis' test

probability $p_{kw}$ are shown instead of $R^2$. Data belonging to the second year of observations are represented as green squares (n = 15, n = 12 for the ratio $NO_y \cdot CO^{-1}$).Error bars associated to the measurement of 1, 10, 100, 1000 $INPs_{-8} \cdot ml^{-1}$ are represented close to the graphs for clarity."

Concerning the classified data, these are the numbers of data per class for year 1:

- Season: $n_1 = 29$; $n_2 = 25$; $n_3 = 11$; $n_4 = 19$
- Source region: $n_1 = 29$; $n_2 = 32$; $n_3 = 23$

The presence of unbalanced groups seems to be more relevant for season, but in the opposite direction as indicated. The seasons with more $INPs_{-8}$ were those with fewer measurements. To cover the point of unbalanced groups, we conservatively substituted the $R^2$ value with the probability associated with Kruskall-Wallis non-parametric test for comparison among groups, which considers the presence of different numbers of data in the groups. Despite this, its results do not differ much from other statistical tests. Coherently, this test has been introduced in section 2.3, page 4 line29: "For the categorical parameters "season score" and "source region score" the results of parametric regression were conservatively substituted with Kruskall-Wallis non-parametric test for the comparison among groups, because this test takes into account the presence of different numbers of data among groups."

Given the absence of a significant change in the statistical results and with the aim of keeping the figure and the figure caption as clear as possible, we would rather not indicate the number of data per group in the figure.

**(18)** *Figure 2/3: suggest adjusting the symbols such that the paper can be printed and understood in black and white.*

Symbols have been changed to facilitate reading in black and white. The figure has been split into 2 separate figures, to improve readability. The captions of the figures have been changed accordingly, as well as the references to them in the main text body.

**(19)** *Figure 3: This figure isn't really discussed in the text. I would consider either expanding its presence in the text or removing it.*

Figure removed.

***Technical Comments:***

**(20)** *Page 3, line27. Suggest insert "of" before CO, and changing "values" to concentrations, and deleting "the concentrations of" from the middle of the sentence: "...hourly concentrations of CO and total reactive...".*

Done, now the sentence reads as: "…Empa also provided hourly concentrations of CO and total reactive nitrogen $NO_y$ in the air."

**(21)** *Page 4, line 14: insert a comma after "season score".*

Done.

**(22)** *Page 5, line 41: Citation for Wright et al is misspelt.*

Corrected the spelling.

**(23)** *Page 6, line 33: Delete "the dots for".*

Done.

**(24)** *Page 7, line 15: Please re-write the sentence so that the paragraph does not start with the word "Thus".*

Substituted "Thus, …" with "As a consequence, …".

**(25)** *Page 7, line 18: Delete ", also for validation".*

The whole sentence starting at page 7, line 17, has been modified into: "Model 1, based only on the two parameters $f_V$ and wind speed, provided better results in predicting the variability of INPs$_{-8}$ observed during the second year of sampling, producing lower absolute errors, less than one log unit (Table 2)."

**(26)** *Page 7, line 37: "DeMott", not "De Mott" (please check elsewhere as well)*

Done and checked elsewhere.

**References mentioned in the reply**

Després, V. R., Huffman, J. A., Burrows, S. M., Hoose, C., Safatov, A. S., Buryak., G., Fröhlich-Nowoisky, J., Elbert, W., Andreae, M. O., Pöschl, U., and Jaenicke, R.: Primary biological aerosols particles in the atmosphere: a review, Tellus B, 64, doi.org/10.3402/tellusb.v64i0.15598, 2012.

Stopelli, E., Conen, F., Zimmermann, L., Alewell, C., and Morris, C. E.: Freezing nucleation apparatus puts new slant on study of biological ice nucleators in precipitation, Atmos. Meas. Tech., 7, 129–134, doi.org/10.5194/amt-7-129-2014, 2014.

Stopelli, E., Conen, F., Morris, C. E., Herrmann, E., Bukowiecki, N., and Alewell, C.: Ice nucleation active particles are efficiently removed by precipitating clouds, Sci. Rep., 5, 16433, doi.org/10.1038/srep16433, 2015.

**Referee #2:**

*Stopelli et al. analyze a two year time series of ice nucleating particles collected from falling precipitation at the Jungfraujoch high altitude observatory. The data are analyzed using simple correlation and multi-linear regression analysis. I read the paper with great interest. Understanding atmospheric variability of 'warm' ice nucleating particles is important and the factors that control their abundance are poorly understood. This paper presents some new insights on the subject.*

*However, the multi-linear regression models do not help the community to glean widely generalizable conclusions. They are location specific. Their ability to explain the temporal variation in the data does not rise beyond an order of magnitude estimate.*

*The conclusions are overstated. This work shows that there is some statistical correlation between INP @-8C in precipitation and temperature, fv, wind speed, source region, N500, and source region. Understanding the uncertainties in the data, the physical mechanisms driving these correlations, and the covariance matrix between the identified parameters is needed. Any claim about the role of INP in precipitation and impact on the water cycle is not supported by this work needs to be removed from the paper. The authors need to more clearly articulate the utility and limitations of the proposed statistical models. Notwithstanding a report on the inferred correlations merits publication to guide the experimental design of future studies.*

*Comments:*

**(1)** *The dynamic range of the measured signal log(INP @ -8C) is 2.5 orders of magnitude, ranging from -0.5 to 2 in Fig. 2. The majority of data fluctuate between -0.5 and 0.5, and so do the residuals of the model. The statistical model is successful in identifying the high INP episodes Jun/Aug 2013 and May 2014. Other fluctuations seem to be mostly noise in either the data, the model, or both. I have several question regarding Fig. 2 and the model. What is the uncertainty in the concentration (temperature) data? Were frequent pure water tests run to determine the measurement noise?*

Blanks were periodically prepared and analysed to test the quality of our sampling and analyses. Out of 29 blanks analysed during the campaigns mentioned in the paper, only 2 of them presented some activity at -8 °C (0.11 INPs$_{-8}$·ml$^{-1}$), indicating that only very low concentrations of INPs could be influenced by our method of analysis. More detail on the methodology and on the blank preparation has been added at the end of section 2.1: "Snow samples were analysed for the concentrations of INPs$_{-8}$ directly on site, using the automated drop freeze apparatus LINDA loaded with 52 tubes containing 100 µl of sample each (prepared adding 2 ml of 9 % NaCl sterile solution to 18 ml of sample and gently shaken, to ensure a final physiological saline concentration and improve the detection of freezing events, dilution 1:1.1, Stopelli et al. 2014; Stopelli et al., 2015). Blanks were periodically prepared distributing sterile Milli-Q water onto the rinsed tin and analysed with the same procedure as the snow samples, with 200 µl per tube to obtain more restrictive results. Out of 29 blanks analysed during the sampling campaigns reported here, only 2 blanks contained 0.11 INPs$_{-8}$·ml$^{-1}$, confirming the accuracy of our analyses."

Further details on errors associated to measurements are provided in comment 2 Referee #2 and comment 16 Referee #1. We also think that some of the fluctuations for low values depend on the fact that the error related to counting frozen tubes is larger for low counts.

**(2)** *INP spectra can be steep functions of temperature. How do small errors in temperature propagate into INP @-8 °C?*

The precision of the measurements with LINDA apparatus is better than 0.2 °C (Stopelli et al., 2014). If the number of INPs active at a certain temperature doubles with each 1 °C decrease, then the errors associated to the determination of $INPs_{-8}$ range between +15 % and -13 % (corresponding to the numbers of INPs active at -8.2 and -7.8 °C respectively). Such values have been combined with the error associated to counting frozen tubes. This is described in a new paragraph is section 2.1 and errors have been added as reference close to figures of the article, as suggested in comment 16 Referee #1.

**(3)** *It seems that the calibration time-series for the model is at higher resolution than the data? (Trying to count green and black dots and get different numbers).*

The resolution of observed and modelled data is the same. We hope this is clearer in the revised version of the manuscript with the improved quality of the figures according to comment 18 Referee #1.

**(4)** *How does measurement noise, in principle, affect the multi linear regression analysis? Residuals can be interpreted as either measurement error or insufficient number or assumed physical relationship of the proposed explanatory variables. How can these be separated?*

Errors associated with measurements have been discussed in point 1 and 2 Referee #2 and point 16 Referee #1. Their relative magnitude is almost constant over several orders of magnitude, which means that errors are equally distributed on all measurements. Also residuals present no specific trend. No physical relationship has been assumed. Concerning "*insufficient number*": if it is related to the number of samples, the dataset employed seems sufficient to us; if it is related to low numbers of $INPs_{-8} \cdot ml^{-1}$, this has been treated at point 6 Referee #1. We agree that in models residuals can contain multiple factors, which cannot be easily separated.

**(5)** *The highest INP were observed for very high wind speeds (60-80 km $h^{-1}$). Were precautions taken that resuspended snow and wind-blown aerosol did not contaminate the measurements?*

We have several hints to exclude the fact that high wind speed massively impacted our measurements by re-suspending snow. Firstly, the tin we used to sample precipitation was placed horizontally on the terrace of the Observatory to maximize the recovery of falling snowflakes instead of those floating across the terrace with wind. Secondly, in a recent paper (Lloyd et al., 2015) it is indicated that snow blown from the surfaces surrounding the

observatory plays a role when ice particles (IPs) have small concentrations and such surfaces can be a source of secondary IPs. In our case we measure INPs by immersion freezing and not IPs or secondary IPs, moreover we observe the opposite behavior with more INPs in association with high wind speed. Thirdly, may it be that some wind-blown INPs get collected, these should have been freshly deposited on the snow surfaces around the station by the same air masses originating precipitation above the station, which makes them not much different from the one we collect on the terrace of the observatory.

**(6)** *Even if the source is local, how could local vs. long-range sources be distinguished?*

This is an interesting question, which cannot be entirely disentangled by measurements done at a single observatory. We tried to obtain some information on the remoteness of sources using the $NO_y \cdot CO^{-1}$ ratio and the source sensitivity plots calculated with FLEXPART model in backward mode. In the revised version of the manuscript, we indicate that under high wind speeds particles from more local sources may be easily uplifted, based on the correlation among wind speed, $N_{>0.5}$ and $NO_y \cdot CO^{-1}$ (comment 9 Referee #1). Concerning the potential impact of wind-blown ice crystals, we conceive it as negligible (please refer to the answer to comment 5 Referee #2). A precise quantification of how much a specific source of particles contributed to the $INPs_{-8}$ measured at Jungfraujoch and how far it was from the station is still a goal to be reached. Ideas on potential ways to tackle this issue in future studies are appreciated.

**(7)** *It is difficult to foresee all the possible cross-correlations. One of the strongest predictors for high INP seems to be temperature (even though the R^2 is low). Values of log(INP) > 1 only under the condition that temperature is at 0 °C. Thus the ability of the model to guess the high INP events is to guess when it is warm, which correlates with season, source region, RH, N>500. Assigning explanatory power to any of the variables in terms of a physical mechanism is likely erroneous.*

In the article, no precise and detailed physical mechanism is attributed to any of the variables. On the contrary, we tried to understand which variables are best correlated with $INPs_{-8}$ and, looking at correlations among variables and at the 3 best models, we tried to find possible driving forces for the variability of INPs. We propose the idea that $INPs_{-8}$ are controlled by source and sink driving forces and that the best proxies for such "black box" mechanisms are $f_V$ and wind speed. This leads to thinking that the frontal passage coincides with the best conditions where to expect large numbers of $INPs_{-8}$. Please refer to the following point 8 Referee #2 for better explanations.

Concerning temperature, it could be a good proxy of "sink" mechanisms. Still, high-INP events do not always coincide with warm air masses. Over the second year of observation exactly the contrary has been registered. We hope that the addition of these points in figure 1 as green squares helps clarifying this (comment 17 Referee #1). The fact that around 0 °C all values of $INPs_{-8}$ are possible doesn't make of temperature an ideal descriptor of their abundance. Therefore, temperature is an important parameter but too locally constrained and does not provide any information on the precipitation history of an air mass, which seems crucial.

To comment on the threshold trend found for temperature, line 27 on page 5 in the text has been rearranged and expanded: "A better linear fit suggests that $f_V$ is a factor capable of better

representing the temporal variability in INPs$_{-8}$ than air temperature, which shows a threshold trend. Specifically, it is possible to find more than 10 INPs$_{-8}$·ml$^{-1}$ in precipitation for temperatures around 0 °C, indicating that when at the station the temperature is warm then also the temperature of precipitation formation in clouds above the Station can be compatible with residual large abundance of INPs$_{-8}$, but not exclusively associated only to large values of INPs$_{-8}$. Therefore, whilst air temperature appears more like a local snapshot-value for the potential activation of INPs$_{-8}$, $f_V$ is a broader descriptor of the cumulative precipitation history of an air mass".

**(8)** *It is still unclear to this referee what the potential application of the statistical would be. A parameterization for a model? Improved process level understanding? Neither approach would seem promising to me without much more analysis. Perhaps the authors have another use in mind?*

Thank you for this point, which allows us to better clarify the meaning of our work. There is large variability in the numbers of INPs in precipitation, specifically for those active at moderate supercooling (Petters and Wright, 2015). With our work we want to understand which factors are responsible for such variability and point at the best conditions where and when to expect large abundance of INPs$_{-8}$.

Here we built multiple linear regression models combining parameters which a priori were already suspected to impact the abundance of INPs$_{-8}$ in precipitation based on previous observations and studies. This can help in assessing the role of such parameters in connection with INPs$_{-8}$. Results are certainly site specific, but this just means that the coefficients of the linear regression are likely to be site specific. There are important more general results which emerge from the comparison of the 3 best models:

- Wind speed is a necessary component in all 3 models and has a relation with $N_{>0.5}$ and $NO_y \cdot CO^{-1}$ (Table 1);
- Combining other parameters with wind speed, 3 models with a similar predictive power were created;
- $f_V$, temperature, season, source region, $N_{>0.5}$ are these further parameters and they are also well correlated among themselves (Table 1), indicating that they are all connected to the same process. It is therefore easy to link them to the processing of particles in the atmosphere. In this perspective, season and source region seem to relate to the potential of an air mass to reach Jungfraujoch with prior little precipitation;

These results point at the more generalizable fact that two driving forces are responsible for the large variability of the abundance of INPs$_{-8}$ in precipitation: the "sink" of particles, to which $f_V$, temperature, season, source region, $N_{>0.5}$ are related, and the "source" of particles, which seems to be linked to wind speed and partly to $N_{>0.5}$. The most synthetic and best performing model including these two driving forces is the one based on $f_V$ and wind speed. Considering these results, we can generalize that the best condition to find large numbers of INPs$_{-8}$ are represented by the passage of a front, where meteorological conditions are also favourable to the onset of precipitation. This also indicates the best conditions where to conduct future measurements.

To clarify these concepts, several modifications have been made to the manuscript:

- page 2 line 24 the sentence now reads: "Prediction of the quantity of INPs$_{-8}$ provides useful means to understand the factors responsible for their large variability in precipitation (Petters and Wright, 2015) and to indicate the circumstances when and where INPs$_{-8}$ may

be sufficiently abundant to impact the formation of the ice phase in clouds and conduce to precipitation."

- page 7 line 33 section title changed into: "3.3 Source and sink effects"
- the whole section 3.4 (now 3.3) has been modified and now reads: "Even if the linear coefficients are site-specific, the 3 models presented in this paper point at important general indications. Wind speed is a necessary parameter to describe INPs$_{-8}$ in precipitation and is related to N$_{>0.5}$ and to a more recent land contact represented by the ratio NO$_y$·CO$^{-1}$ (table 1). Other factors can be combined to wind speed obtaining models of comparable quality. These factors are $f_V$, temperature, season, source region, and N$_{>0.5}$ and are all well correlated among themselves (table 1). This means that their relation with INPs$_{-8}$ can be linked to the same process, which we suspect to be particle processing in precipitating clouds. This can act as "sink" force for INPs$_{-8}$ and is best represented by $f_V$. On the other hand, wind speed can strengthen the "source" of airborne particles. To different degrees, surfaces on Earth, like oceans, forests, crops, soils, freshwaters, snow packs, host organisms with ice nucleating activity (Després et al., 2012) and may contribute to the airborne population of INPs$_{-8}$. In this perspective, source region and seasonality may relate more to the likelihood for an air mass to reach Jungfraujoch with a lot of particles and little prior precipitation rather than to a different background number of airborne INPs$_{-8}$. Therefore it is possible to imagine that, independent from a more or less constant and widespread reservoir of INPs$_{-8}$, it is the combination of the energy of an air mass with the amount of precipitation generated by this air mass that determines the residual abundance of INPs$_{-8}$ in precipitation samples."
- Please refer to comment 14 Referee #1 for changes in the conclusion.

**(9)** *Relating INP in precipitation to INP in air, and INP in air feeding the cloud, is problematic due to a wide range of known artifacts. These include capture of INP by falling rain snow and chemical or physical modification of INP by the precipitation. None of these issues are acknowledged in the manuscript. Even if the explanatory power of the model were perfect and the intrinsic measurement error of LINDA was zero, what relevance do INP @-8 °C measured in precipitation have to clouds over Jungfraujoch?*

Please refer to the answer to comment 1 Referee #1.

**(10)** *The discussion in Section 3.3 should be removed since they discuss very specific cloud types, updrafts and meteorological conditions. It is entirely unclear to what extent the INP @-8 °C at Jungfraujoch have any role in precipitation without considering the larger scale meteorological forcing, including liquid water content, cloud top height, updraft velocity, the INP spectrum at warmer and colder temperatures, the efficiency of the competing warm rain process, and the cumulative ice phase history, including possible seeding by wind-blown ice crystals along the storm trajectory, including all mountains upwind. No information can be gleaned from this work about any of these processes.*

Our idea was not to infer to what extent INPs$_{-8}$ have a role on precipitation, but just to indicate when they may have been largely abundant in the atmosphere, according to available literature reporting aircraft observations. Nevertheless, based on information collected at Jungfraujoch we cannot resolve with such proposed level of detail all the processes happening in clouds. Therefore the former section 3.3 has been removed from the manuscript. Coherently, also the sentence on page 8 lines 12-14 has been removed.

**References mentioned in the reply**

Lloyd, G., Choularton, T. W., Bower, K. N., Gallagher, M. W., Connolly, P. J., Flynn, M., Farrington, R., Crosier, J., Schlenczek, O., Fugal, J., and Henneberger, J.: The origins of ice crystals measured in mixed-phase clouds at the high-alpine site Jungfraujoch, Atmos. Chem. Phys., 15, 12953-12969, doi:10.5194/acp-15-12953-2015, 2015.

Petters, M. D., and Wright, T. P.: Revisiting ice nucleation from precipitation samples, Geophys. Res. Lett., 42, 8758-8766, doi.org/10.1002/2015GL065733, 2015.

Stopelli, E., Conen, F., Zimmermann, L., Alewell, C., and Morris, C. E.: Freezing nucleation apparatus puts new slant on study of biological ice nucleators in precipitation, Atmos. Meas. Tech., 7, 129–134, doi.org/10.5194/amt-7-129-2014, 2014.

**Additional insertions and improvements**

(1) The sentence on page 5 line 3 now reads as: "The normal distribution of independent and dependent variables is considered as not necessary for assessing the quality of multiple linear regression models, but it can improve the quality of the results of the model."

(2) Inserted the reference "Bowers et al., 2009" in page 2 line 9, page 5 line 41 and in the list of references.

(3) Moved the reference "Pöschl et al., 2009" to page 2 line 4.

(4) Moved the reference "Lindow et al., 1978" to page 4 line 17.

(5) Moved the reference "Huffman et al., 2013" to page 2 line 5.

(6) Added at the end of acknowledgements section: "Trace gases and $PM_{10}$ measurements at Jungfraujoch are performed as part of the Swiss National Air Pollution Monitoring Network which is operated by Empa in collaboration with the Swiss Federal Office for the Environment."

(7) Substituted in the acknowledgement section: "by the European FP7 project BACCHUS (grant agreement no. 49603445)." with "by the European FP7 project BACCHUS (grant agreement no. 603445)."

(8) Changed part of the sentence on page 5 line 23 which now reads as: "…our current understanding of the factors that are related to the abundance of INPs…"

[revised manuscript text omitted]
 understand the factors responsible for their large variability in precipitation (Petters and Wright, 2015) and to indicate the circumstances when and where $INPs_{-8}$ may be sufficiently abundant to impact the formation of the ice phase in clouds and conduce to precipitation.

**2 Methods**

**2.1 Sample collection and analysis of INPs**

Falling snow was collected at the High Altitude Research Station Jungfraujoch in the Swiss Alps (7°59'06'' E, 46°32'51'' N, 3580 m a.s.l.) from December 2012 until October 2014. A total of 106 precipitation samples were
35 collected over these two years, with a median sampling duration of about 2 hours per sample (sampling time between 1.5 and 8 hours), depending on the intensity of the precipitation events. We started sampling campaigns when the forecasts predicted 2 or more days of precipitation to assure the collection of several samples during each campaign. Samples were collected with a Teflon-coated tin carefully rinsed with ethanol and sterile Milli-Q water to avoid cross-contamination.
40 Snow samples were analysed for the concentrations of $INPs_{-8}$ directly on site, using the automated drop freeze apparatus LINDA loaded with 52 tubes containing 100 µl of sample each (prepared adding 2 ml of 9 % NaCl

sterile solution to 18 ml of sample and gently shaken, to ensure a final physiological saline concentration and improve the detection of freezing events, dilution 1:1.1, Stopelli et al. 2014; Stopelli et al., 2015). Blanks were periodically prepared distributing sterile Milli-Q water onto the rinsed tin and analysed with the same procedure as the snow samples, with 200 µl per tube to obtain more restrictive results. Out of 29 blanks analysed during the sampling campaigns reported here, only 2 blanks contained 0.11 INPs$_{-8}$·ml$^{-1}$, confirming the accuracy of our analyses.

Error bars for values of 1, 10, 100, 1000 INPs$_{-8}$·ml$^{-1}$ are shown in the figures of the article. Confidence intervals were calculated as errors in counting frozen tubes, following Poisson's distribution (depending on the number of frozen tubes, they account for 30 % to 50 % increase or decrease of the calculated concentrations of INPs$_{-8}$). These intervals were propagated into the uncertainty associated with the maximum error in the determination of the freezing temperature of the tubes of ± 0.2 °C (assuming a doubling of INPs per °C of decrease in the freezing temperature, an error of 0.2 °C accounts for a change in 14 % of the measured concentrations) to provide more cautious confidence intervals.

The collection of precipitation allows the sampling of INPs which either formed precipitating ice particles or were scavenged by precipitation. It is difficult to distinguish between these contributions in field studies, where scavenging, riming and crystal growth by vapour deposition can alter the abundance of INPs in precipitation. Nevertheless, the station was always inside clouds while sampling, allowing us to collect falling snow as close as possible to where it formed. Furthermore, precipitation was immediately analysed, in order to minimise the chance for biases due to artefacts like the production (f.i. the release in solution of INPs or cellular multiplication) and the loss (f.i. settling or increased molecular weakness of biological INPs detached from mineral and soil dust) of INPs$_{-8}$

**2.2 Parameters related to the concentration of INPs**

To analyse and understand more on the factors responsible for the variability of INPs$_{-8}$ in precipitation, several environmental parameters were considered in relation with the number of INPs$_{-8}$.

[revised manuscript text omitted]

The normal distribution of independent and dependent variables is considered as not necessary for assessing the quality of multiple linear regression models, but it can improve the quality of the results of the model. Consequently, the variables which were log-transformed for univariate statistics were kept transformed also in multiple linear models. The quality of a multiple linear regression model is evaluated by the significance of the whole model as well as of the regression coefficients of each parameter. Particular care was taken in analysing the residuals of the models. All the models presented here fulfilled the conditions of normally distributed residuals, with an average value of zero and no significant trends. Furthermore, we assumed that the parameters could be added in linear combinations. The correctness of this assumption was verified by the method of partial regression plots of the residuals. Given an ideal model y ~ x$_1$ + x$_2$ (x$_1$ and x$_2$ used to derive the dependent variable y) it is possible to test whether f.i. x$_2$ is linearly linked to y. To do that, the residuals of the regression of y with x$_2$ are plotted against the residuals of the regression of x$_1$ with x$_2$. A linear distribution of the residuals confirms the correctness of the linear relationship between x$_2$ and y. On the contrary, the presence of a different

trend implies a different relationship between $x_2$ and y, like, for example, a quadratic one. Interactions between independent variables were tested as well as potential ways to improve the models. This means that the additional factor $x_1 \cdot x_2$ was inserted in a model to test whether the effect of the independent variable $x_1$ (or $x_2$) on the dependent variable y changes according to different levels of the other independent variable $x_2$ (or $x_1$). No interaction we tested resulted in a significant improvement of the models.

**3 Results and discussion**

**3.1 Model calibration**

The observations used to create the models consist of 84 snow samples with measurable concentrations of INPs$_{-8}$ collected in the Swiss Alps at 3580 m altitude between December 2012 and September 2013. Measured values of INPs$_{-8}$ ranged from the lower limit of detection ($0.21 \cdot ml^{-1}$) to a maximum of $434 \cdot ml^{-1}$. Interestingly, these values are comparable to, or even greater than those recently found in cloud water samples in Central France at 1465 m altitude (Joly et al., 2014) and well within the range of values and variability observed in precipitation samples collected all around the world (Petters and Wright, 2015).

The best correlations found at Jungfraujoch agree with our current understanding of the factors that are related to the abundance of INPs in the environment (Fig. 1, black dots). In particular, the relationships with the remaining fraction of water vapour $f_V$ and air temperature are coherent with the observation that INPs are rapidly lost by precipitating clouds, hence are more abundant at early stages of precipitation (Stopelli et al., 2015) and that colder air masses tend to be more depleted in INPs$_{-8}$ (Conen et al., 2015). A better linear fit suggests that $f_V$ is a factor capable of better representing the temporal variability in INPs$_{-8}$ than air temperature, which shows a threshold trend. Specifically, it is possible to find more than 10 INPs$_{-8} \cdot ml^{-1}$ in precipitation for temperatures around 0 °C, indicating that when at the station the temperature is warm then also the temperature of precipitation formation in clouds above the Station can be compatible with residual large abundance of INPs$_{-8}$, but not exclusively associated only to large values of INPs$_{-8}$. Therefore, whilst air temperature appears more like a local snapshot-value for the potential activation of INPs$_{-8}$, $f_V$ is a broader descriptor of the cumulative precipitation history of an air mass.

[revised manuscript text omitted]

15 As a consequence, models 2 and 3, which are based either on season, source region or air temperature, predicted a smaller variability of INPs than observed and overestimated the low concentrations measured in summer 2014 causing larger residual values (Fig. 3 lower panel, Table 2). Model 1, based only on the two parameters $f_V$ and wind speed, provided better results in predicting the variability of INPs$_{-8}$ observed during the second year of sampling, producing lower absolute errors, less than one log unit (Table 2).

**3.3 Source and sink effects**

Even if the linear coefficients are site-specific, the 3 models presented in this paper point at important general indications. Wind speed is a necessary parameter to describe INPs$_{-8}$ in precipitation and is related to $N_{>0.5}$ and to

25 a more recent land contact represented by the ratio $NO_y \cdot CO^{-1}$ (table 1). Other factors can be combined to wind speed obtaining models of comparable quality. These factors are $f_V$, temperature, season, source region, and $N_{>0.5}$ and are all well correlated among themselves (table 1). This means that their relation with INPs$_{-8}$ can be linked to the same process, which can be particle processing in precipitating clouds. This can act as "sink" force for INPs$_{-8}$ and is best represented by $f_V$. On the other hand, wind speed can strengthen the "source" of airborne

30 particles. To different degrees, surfaces on Earth, like oceans, forests, crops, soils, freshwaters, snow packs, host organisms with ice nucleating activity (Després et al., 2012) and may contribute to the airborne population of INPs$_{-8}$. In this perspective, source region and seasonality may relate more to the likelihood for an air mass to reach Jungfraujoch with a lot of particles and little prior precipitation rather than to a different background number of airborne INPs$_{-8}$. Therefore it is possible to imagine that, independent from a more or less constant and

35 widespread reservoir of INPs$_{-8}$, it is the combination of the energy of an air mass with the amount of precipitation generated by this air mass that determines the residual abundance of INPs$_{-8}$ in precipitation samples.

40 **4 Conclusions**

Large variability in the abundance of INPs$_{-8}$ has been found in over a hundred precipitation samples collected at Jungfraujoch (3580 m a.s.l.), with values ranging from 0.21 INPs$_{-8}$·ml$^{-1}$ to 434 INPs$_{-8}$·ml$^{-1}$. Strikingly, with simple multiple linear regression models based on some easily measurable environmental parameters it was possible to describe and predict up to 75 % of the observed variability in INPs$_{-8}$. All these models indicate that the variability of INPs$_{-8}$ is determined by the interaction of "source" and "sink" processes, such as the potential for air masses to pick up and transport INPs$_{-8}$ from several sources and for INPs$_{-8}$ to be removed by precipitation. Our investigation indicates that a large abundance of INPs$_{-8}$ in precipitation at Jungfraujoch is present whenever there is a coincidence of high wind speed and moist air mass with little or no prior precipitation. Based on the results of the present study, INPs active at moderate supercooling are expected to be abundant whenever high wind speed coincides with first (initial) precipitation from an air mass. These conditions can be met when an air mass is suddenly forced to rise, for instance at the boundary of a front or due to thermal updrafts or when crossing a mountain ridge. Specifically during the passage of a cold front, gusty winds promote the uptake of particles and the first clouds that form will still retain a large fraction of the initial water vapour of the warm air mass (Gayet et al., 2009; Wright et al., 2014). Simultaneously, physical conditions along a cold front are favourable for cloud formation. Therefore, frequent systematic coincidences of high numbers of INPs with meteorological conditions conducive to precipitation may be expected. Due to this frequent co-occurrence, the potential impact on precipitation by INPs active at slight supercooling - such as INPs of biological origin - may be larger than previously estimated. Their role in the water cycle might therefore best be studied under such conditions.

**Author contributions**

E.S. and F.C. did the field measurements at Jungfraujoch on the concentrations of INPs, analysed data, did statistical modelling and wrote the manuscript. E.H., S.H. and M.S. respectively provided data and support for the interpretation of $N_{>0.5}$, FLEXPART modelling, $PM_{10}$ and trace gases and contributed to writing the paper. C.M. and C.A. provided strong conceptual frameworks and contributed to writing the paper.

**Acknowledgements**

We thank the International Foundation for High Alpine Research Station Jungfraujoch and Gornergrat (HFSJG) for making it possible for us to conduct our measurements at Jungfraujoch. Urs and Maria Otz, Martin and Joan Fischer provided helpful support during field activity. Corinne Baudinot measured the abundance of INPs in snow samples during the second year of observations. Dr Thomas Kuhn and Mark Rollog analysed the stable isotope ratio in our snow water samples. We thank MeteoSwiss for providing data on meteorology at Jungfraujoch. The work described here was supported by the Swiss National Science Foundation (SNF) through grant no 200021_140228 and 200020_159194. Measurements of total solid particles were performed by Paul Scherrer Institute in the framework of the Global Atmospheric Watch (GAW) programme funded by MeteoSwiss with further support provided by the European FP7 project BACCHUS (grant agreement no. 603445). Trace gases and $PM_{10}$ measurements at Jungfraujoch are performed as part of the Swiss National Air Pollution Monitoring Network which is operated by Empa in collaboration with the Swiss Federal Office for the Environment.

The authors declare no competing financial interests.

[revised manuscript text omitted]

 wind speed: **0.47** | Med: -0.04 (0.9)
 ABS: 1.02 (10.4)
 MSE: 0.16 | Med: -0.08 (0.8)
 ABS: 0.87 (7.4)
 MSE: 0.25 |
| 2 | **0.73** | season: **0.52**
 wind speed: **0.52**
 source region: **0.12** | Med: 0.00 (1.0)
 ABS: 1.45 (28.4)
 MSE: 0.17 | Med: -0.25 (0.6)
 ABS: 1.42 (26.1)
 MSE: 0.56 |
| 3 | **0.74** | wind speed: **0.49**
 temperature: **0.48**
 log ($N_{>0.5} \cdot m^{-3}$): **0.26** | Med: -0.02 (0.9)
 ABS: 1.22 (16.5)
 MSE: 0.17 | Med: -0.33 (0.5)
 ABS: 1.13 (13.5)
 MSE: 0.43 |

[Figure]

[Figure]

**Fig. 1:** Relationships between INPs$_{-8}$·ml$^{-1}$ and singular environmental parameters for the first year of observations (black dots, n = 84, exception for the ratio NO$_y$·CO$^{-1}$ with n = 51 due to missing data of trace gases). Shown are R$^2$ values, the coefficient of Spearman's correlation r$_s$ and its probability p$_s$ calculated for the data belonging to the first year of observations. $f_V$ indicates the remaining fraction of water vapour in a precipitating air mass. In the panel "season score" 1 is for winter, 2 for spring, 3 for autumn and 4 for summer. In the panel "source region score" 1 is for Northern Europe, 2 for mixed conditions and 3 for Southern Europe. For both "season score" and "source region", values of Kruskall-Wallis' test probability p$_{kw}$ are shown instead of R$^2$. Data belonging to the second year of observations are represented as green squares (n = 15, n = 12 for the ratio NO$_y$·CO$^{-1}$). Error bars associated to the measurement of 1, 10, 100, 1000 INPs$_{-8}$·ml$^{-1}$ are represented close to the graphs.

[Figure]

[Figure]

**Fig. 2:** Comparison of observed concentrations of INPs$_{-8}$ with the values predicted by models (absolute values in upper panels, residuals in lower panels) for the dataset used for calibration (n = 84). The grey area in the upper panel indicates the range below the detection limit of our observation method. Time proceeds from left to right,

5    intervals are not to scale, dots belonging to the same sampling campaign are connected by lines. Error bars associated to the measurement of 1, 10, 100, 1000 INPs$_{-8}$·ml$^{-1}$ are represented close to the graph.

[Figure]

**Fig. 3:** Comparison of observed concentrations of INPs$_{-8}$ with the values predicted by models (absolute values in upper panels, residuals in lower panels) for the dataset used for validation (n = 15). The grey area in the upper panel indicates the range below the detection limit of our observation method. Time proceeds from left to right, intervals are not to scale, dots belonging to the same sampling campaign are connected by lines. Error bars associated to the measurement of 1, 10, 100, 1000 INPs$_{-8}$·ml$^{-1}$ are represented close to the graph.

[Figure]

**Fig. 3:** Fitting of modelled concentrations of INPs$_{-8}$ to the observed values, expressed on the log scale. The black line represents the perfect 1:1 correspondence. Specifically for the validation, it is possible to observe how model 2 and 3 tend to overestimate the lowest concentrations of INPs$_{-8}$ measured during summer 2014.